# ¡Miranos! An 8-Month Comprehensive Preschool Obesity Prevention Program in Low-Income Latino Children: Effects on Children’s Gross Motor Development

**DOI:** 10.3390/ijerph20216974

**Published:** 2023-10-25

**Authors:** Vanessa L. Errisuriz, Deborah Parra-Medina, Yuanyuan Liang, Jeffrey T. Howard, Shiyu Li, Erica Sosa, Sarah L. Ullevig, Vanessa M. Estrada-Coats, Zenong Yin

**Affiliations:** 1Department of Public Health, Santa Clara University, 500 El Camino Real, Santa Clara, CA 95053, USA; verrisuriz@scu.edu; 2Latino Research Institute, University of Texas at Austin, 210 W. 24th Street, Austin, TX 78712, USA; 3Department of Epidemiology and Public Health, University of Maryland School of Medicine, 660 W. Redwood Street, Baltimore, MD 21201, USA; yliang@som.umaryland.edu; 4Department of Public Health, University of Texas at San Antonio, One UTSA Circle, San Antonio, TX 78249, USA; jeffrey.howard@utsa.edu (J.T.H.); erica.sosa@utsa.edu (E.S.); vanessa.estrada@utsa.edu (V.M.E.-C.); zenong.yin@utsa.edu (Z.Y.); 5School of Nursing, UT Health San Antonio, 7703 Floyd Curl Dr, San Antonio, TX 78229, USA; lis9@livemail.uthscsa.edu; 6Department of Nutrition and Dietetics, University of Texas at San Antonio, One UTSA Circle, San Antonio, TX 78249, USA; sarah.ullevig@utsa.edu

**Keywords:** preschool children, gross motor development, Latino, low income

## Abstract

Organized childcare is an ideal setting to promote gross motor development in young children from low-income minority families. A three-group clustered randomized controlled trial was conducted in Head Start centers serving low-income Latino children to evaluate the impact of an 8-month comprehensive obesity-prevention intervention on children’s percentile scores for locomotive skills (LS pctl) and ball skills (BS pctl), and general motor quotient (GMQ). Trained Head Start staff delivered the center-based intervention (CBI) to modify center physical activity and nutrition policies, staff practices, and child behaviors, while the home-based intervention (HBI) offered training and support to parents for obesity prevention at home. Participants were 3-year-old children (*n* = 310; 87% Latino; 58% female) enrolled in Head Start centers in South Texas. Twelve centers were randomized (1:1:1 ratio) to receive CBI, CBI and HBI (CBI + HBI), or control treatment. Posttest data were collected from 79.1% of participants. All gross motor development measures improved significantly for children in CBI compared to the control, while children in CBI + HBI only showed improvement for GMQ (*p* = 0.09) and LS pctl (*p* < 0.001) compared to the control. A comprehensive and culturally competent intervention targeting childcare centers and children’s homes was effective at improving children’s gross motor development and reducing disparities in child development.

## 1. Introduction

Gross motor skills (GMS), also known as fundamental motor or movement skills, comprise stability (e.g., balancing), object control (e.g., ball throwing), and locomotor (e.g., running) movements [1] and play a critical role in children’s growth and development [2]. Many GMS develop during preschool years (3–5 years old) and provide a foundation for children to develop more specialized movement sequences for sport-specific and lifelong activity [3,4]. Children’s gross motor competence, that is, their ability to perform different fine and gross motor acts required to manage everyday tasks [5], is associated with a variety of health behaviors and outcomes, including physical activity, fitness, and weight status. For instance, a low level of gross motor competence is associated with less physical activity and poorer cardiorespiratory fitness and weight outcomes among children [6,7,8]. Evidence from longitudinal research indicates that children’s gross motor competence has long-term effects on health as low competence is associated with a decline in physical activity levels and fitness as well as increased body mass index over time [9,10]. In turn, children with overweight or obesity often experience challenges performing motor skills because of their increased mass, which perpetuates physical inactivity throughout childhood [11,12].

Additionally, GMS are linked to improved cognitive and behavioral outcomes among children such as working memory, attention, and prosocial behaviors [11,13,14,15]. GMS and cognitive function are highly interconnected [16], and GMS have long been considered essential for children’s early learning and long-term academic success [17]. Guidelines from national organizations and federal agencies in the U.S.A. recognize the critical role of motor development in promoting children’s social–emotional, cognitive, and language/literacy development in children aged 2–5 and the importance of incorporating motor-skill development in early-childhood education programs [18,19,20].

Despite the benefits associated with gross motor development, levels of GMS among American children are low. One study examining data from the 2012 National Health and Examination Survey National Youth Fitness Survey found that 33% of U.S. children aged 3–5 years scored below average for GMS [21]. Another study found that 51.9% and 57.1% of U.S. preschoolers performed below average on locomotor skills and object control skills, respectively [22]. Latino children and those from low-socioeconomic-status (SES) backgrounds are particularly at risk of GMS delays, with research indicating that Latino preschoolers have a high prevalence of poor object-control skills [23] and that they perform significantly worse on measures of fine and gross motor skills when compared to their age-matched, higher-SES peers [24]. Other research shows that children identified as coming from low SES backgrounds are significantly delayed in both fine and gross motor skill areas when compared to typically developing age- and gender-matched counterparts [25,26,27,28] due to various factors, including limited access to physical activity resources in the home or neighborhood, lack of available play equipment, and insufficient funds to allow for participation is sports or other recreational activities [25].

Hulteen et al. proposed a conceptual model that outlines developmental mechanisms that influence young children’s physical-activity trajectories and highlights the reciprocal relationship between motor-skill competence and physical activity [29]. The model posits that physical activity in early childhood drives development of motor-skill competence such that greater physical activity promotes motor-skill development. In turn, greater motor-skill competence leads children to being more active later in childhood as they will have a larger inventory of motor skills that enables them to engage in a variety of physical activities. Although the relationship between physical activity and motor competence is weak during early childhood, the relationship is thought to strengthen over time due to psychological and physical factors such as perceived motor competence, physical fitness, obesity, and the environment as well as a sociocultural and geographical filter that informs which movement skills are developed [29].

Varying levels of motor competence among young children are primarily influenced by differing experiences, including those related to the environment (e.g., home, preschool, built environment) as it influences the opportunities that children have to be active throughout the day [2]. During preschool years, children spend much of their day (~33 h per week) in out-of-home care, including early childcare centers such as Head Start [30,31]. As such, preschool environments may play a large role in shaping children’s motor development [32]. Indeed, improvements in gross motor skills are intertwined with the quality of the preschool setting [33]. Unfortunately, findings from a large national study found that U.S. preschoolers spent only 6% of classroom time engaging in gross motor activities, defined in this study as any activity that includes large muscles of the arms and/or legs (e.g., running, playing ball, dancing) [34]. Six percent of classroom time is equivalent to almost 2 h of a 33 h week, which is significantly less than the 2 h of physical activity (i.e., 60 min structured and 60 min unstructured) per day that national guidelines suggest.

This article reports findings from ¡Míranos! Look at Us, We Are Healthy! (¡Míranos!), a multi-level obesity-prevention program for low-income Latino children enrolled in Head Start that targeted multiple energy-balance-related behaviors (e.g., diet, physical activity) via a center-based intervention (CBI) and home-based intervention (HBI) designed to reduce barriers and enhance enablers of obesity prevention [35]. The purpose of this study was to examine the effects of the 8-month ¡Míranos! intervention on children’s gross motor development. In line with Hulteen et al.’s conceptual model, changes to the Head Start environment that promote physical activity should have a positive impact on young children’s gross motor competence. Previous research has, in fact, demonstrated that participation in obesity-prevention programs, particularly those with physical-activity components, is associated with improvements in preschool children’s gross motor skills [36,37,38]. Thus, the study hypothesis was that children in the combined CBI and HBI or CBI-only conditions would experience a significantly larger improvement in gross motor skills compared to control children at the post test immediately after the completion of the intervention.

## 2. Materials and Methods

### 2.1. Study Design

¡Míranos! was a three-arm clustered randomized controlled trial conducted at twelve Head Start childcare centers in San Antonio, Texas. Head Start is a federally funded program that offers services in school readiness, health, nutrition, and social services for children from birth to five years old from families that meet the federal low-income guidelines [39]. The research protocol detailing the design and intervention, and results of the effects on the primary and secondary outcomes have been previously reported [35]. This study was conducted in accordance with the Declaration of Helsinki and approved by the Institutional Review Board of the University of Texas at San Antonio (IRB# 18–187).

### 2.2. Study Setting and Participants

To be eligible to participate in the ¡Míranos! RCT, Head Start centers must have had ≥75% of enrolled children identify as Latino, an onsite outdoor playground, serve meals from the center’s central kitchen, agreement to not participate in new health-related research studies during the study period, and agreement to treatment randomization. Children were eligible if they were 3 years old at the beginning of the school year and enrolled in a 3-year-old-only classroom at a participating Head Start center. Only one child per family was included in the study. If more than one child from a family was identified, we only included the first child encountered by data-collection staff. Parents/guardians signed an informed consent for their child to participate in the ¡Míranos! trial. Participants were recruited in two cohorts (Cohort 1: August 2018–May 2020; Cohort 2: August 2019–May 2021). Due to the COVID-19 pandemic, posttest gross motor assessments were not conducted in Cohort 2 participants, who were excluded from this study’s analyses. Participants who had a missing measure of body mass index, the primary outcome measure, were also excluded from this study.

We randomly assigned 12 Head Start centers to the combined center- and home-based intervention (CBI + HBI), the center-based only intervention (CBI), and control condition in a 1:1:1 ratio. The treatment conditions were concealed from the participants until completion of baseline assessments. The investigators and research staff were not blinded to treatment conditions. A full description of the recruitment and randomization process was published previously [35].

### 2.3. Description of ¡Míranos! Intervention

¡Míranos! implemented evidence-based strategies and encouraged best practices to promote key messages targeting Head Start center policies, staff and parent practices, and children’s energy-balance-related behaviors using activities that took place at the center and at home. We briefly describe the three intervention arms here (see Table 1). A thorough description of the development of ¡Míranos! and the three intervention arms was previously published [35,40]. The CBI comprised center policy changes and recommendations for nutrition and meal-pattern modification, enhancement of physical activity and gross motor program, supplemental health-education classroom activities, and a staff wellness program. The HBI consisted of home visits by Head Start family-service workers to help parents set family-health goals, family-health challenges, and obesity-prevention-education sessions for parents led by trained peer educators. During parent-education sessions, parents received take-home bags with obesity-education content, family newsletters, and activities for children to perform at home.

The research team trained intervention center staff (center operators, classroom teachers, teacher assistants, family-service workers, food-service workers, and custodians), central kitchen workers, and senior curricular staff to deliver intervention components. Head Start employees participated in a full day of intervention training and booster training and were compensated for their time. Centers assigned to the control condition implemented the Head Start-endorsed physical activity and nutrition program, “I Am Moving, I Am Learning” [41]. Parents enrolled at control centers were invited to participate in a nutrition-themed literacy-education program supported by a local grocery chain.

### 2.4. Study Measurements

Children’s gross motor development was assessed using a direct observation tool, the Test of Gross Motor Development—2nd edition (TGMD-2), which included a subset of 6 locomotive skills and a subset of 6 ball skills at baseline and post test [42]. The TGMD-2 is a standardized measure of gross motor competence with both criterion (raw score) and nationally normative data for age and sex in children aged 3 to 10 [43,44]. The TGMD-2 was utilized because the Head Start centers enrolled in this study use this measure to test whether children’s gross motor skills meet Head Start Program Performance Standards. Children performed each of the skills twice following a demonstration by a research assistant. The activities were video-taped and subsequently scored by an expert who was blinded to the treatment conditions. For this analysis, we used the general motor quotient (GMQ), and percentiles for locomotive skills (LS pctl) and ball skills (BS pctl) that are based on a presentative national norm for American children. GMS is a composite score of the standard scores of locomotive skills and ball skills that represents overall levels of gross motor development, while LS pctl and BS pctl are rank scores derived from the standard scores [42]. We also collected child and family demographic information and health history from Head Start records and parent surveys.

### 2.5. Statistical Analyses

We examined the demographic characteristics of Head Start centers and study participants in three conditions using descriptive statistics (i.e., Chi-square test or Fisher’s exact test for categorical variables and Kruskal–Wallis H test for continuous variables). For each outcome of interest (i.e., GMQ, LS pctl, and BS pctl), we used a 3-level (time nested within child and child nested within center) linear mixed-effects model (LMM) to examine group differences with time (baseline vs. post test), treatment group (CBI vs. C + HBI vs. control), the interaction between time and treatment group, and center size (large vs. small) as fixed predictors that were kept in the model regardless of whether they were statistically significant. The LMM had one random term to account for the correlation among children nested within the same center and a second term to account for the correlation among two measures nested within the same child. Data were assumed to be missing at random. In the full LMMs, we included child’s age at baseline, age squared, gender, race/ethnicity, asthma, mother’s education, language spoken most often at home, parent marital status, and family history of diabetes as confounders. We employed backward model selection to select the final reduced model by removing one non-significant (*p* > 0.05) confounder at a time, and Akaike’s information criterion (AIC) and the Bayesian information criterion (BIC) guided the model-selection process. The results of the final models are reported. All analyses were performed using Stata/SE (version 16) (StataCorp LLC, College Station, TX, USA).

## 3. Results

Of the 349 Cohort 1 children that consented, 88.8% (*n* = 310) completed baseline assessments and provided valid scores for gross motor development and BMI at baseline (92 C + HBI, 100 CBI, and 118 control), and 79.1% (*n* = 276) were retained at post test (see Figure 1). Children’s mean age was 3.59 (SD = 0.29) years, 58% were female, 87.4% were Latino, 12.9% had a diagnosis of asthma, and 41% had a family history of diabetes. Most mothers had at least a high-school diploma or GED (78%), and more than half of the children spoke English most often at home (55.5%). There were no significant differences in children’s characteristics between the three groups, except that more of the center-based intervention children were from small-sized centers (see Table 2).

Table 3 displays the mean values and standard deviations of GMQ, LS pctl, and BS pctl scores, at baseline and post test across the three intervention groups. Overall, children showed improvement for all gross motor outcomes from baseline to post test in all treatment groups except for BS pctl score in the control group (see Figure 2). At baseline, there was no significant difference in baseline scores between the intervention groups. The posttest and change scores in GMS and LS pctl were higher in CBI + HBI and CBI groups than children in the control group. There were no significant differences between groups for BS pctl scores at post test or for change scores.

The results of the LMM analyses adjusting for outcome-specific significant confounders are summarized in Table 4 and Figure 2. After adjustment, significant within-group increases in GMQ and LS pctl score were observed for all groups. Specifically, children in all three groups, CBI + HBI (mean change (SE) = 8.08 (1.48), *p* < 0.001) CBI (mean change (SE) = 10.15 (1.43), *p* ≤ 0.001), and control (mean change (SE) = 4.69 (1.32), *p* ≤ 0.001) exhibited significant increases in GMQ from baseline to post test. Children also demonstrated significant increases in LS pctl scores from baseline to post test in the CBI + HBI (mean change (SE) = 23.99 (2.48), *p* < 0.001), CBI (mean change (SE) = 17.91 (2.53), *p* < 0.001), and control (mean change (SE) = 10.31 (2.31), *p* < 0.001) groups. However, only children in the CBI group showed a significant increase in BS pctl scores (mean change (SE) = 6.11 (2.52), *p* = 0.015).

Findings from the LMM analyses also showed significant between-group differences (Table 4). Children in the CBI group had significantly greater increases in GMQ from baseline to post intervention compared to children in the control group (5.46 (95% CI: 1.64, 9.29), *p* = 0.01). Children in the CBI + HBI group also experienced a greater increase in GMQ from baseline to post intervention relative to those in the control group; however, this difference did not reach significance (3.39 (95% CI: −0.50, 7.28), *p* = 0.09). Children in the CBI (7.60 (95% CI: 0.89, 14.31), *p* = 0.03) and CBI + HBI (13.68 (95% CI: 7.04, 20.33), *p* < 0.001) groups both demonstrated significantly greater increases in LS pctl from baseline to post test compared to children in the control group. Only children in the CBI group exhibited greater increases in BS pctl scores than their control-group counterparts (7.57 (95% CI: 0.86, 14.28), *p* = 0.03).

## 4. Discussion

¡Míranos! is one of the first intervention studies to report significant effects on gross motor development among Latino children during early childhood, in which movement experience can have a long-term influence on child development [45,46]. In this study, children across all groups (i.e., CBI + HBI, CBI, and control) showed significant improvements in overall gross motor development and locomotive skills, while only the CBI group had a significant increase in ball skills from baseline to post intervention. Compared to the control group, children from both CBI and CBI + HBI groups exhibited larger increases in overall gross motor development and locomotive skills. Although ball skills were improved in both intervention groups, the change was only significant in CBI children compared to control children.

In general, young children are better prepared to develop locomotive skills involving simpler neuromuscular coordination of large muscle group (i.e., body parts). However, ball skills require more complex coordination of vision and muscle groups that are more challenging to master in young children [47]. Therefore, it is more common to see improvement in locomotive skills and overall gross motor development than in ball skills in early childhood [8,48]. However, ¡Míranos! intervention enhanced the development of all three gross motor development measures above and beyond what is observed in the control group, although the improvement in ball skills was not statistically significant in the CBI + HBI group (+4.78 (−2.04, 11.60), *p* < 0.17). These findings are similar to those reported in small pilot studies [49,50,51,52].

The improvements in gross motor skills among children in the intervention groups can be attributed to one of the prominent foci of ¡Míranos! to develop children’s gross motor skills via teacher-led structured physical activities [35], which involved modifications of center policy on structured physical activity and staff training on gross motor development as well as provision of lessons and play equipment designed to teach gross motor skills (see Table 1) [53,54]. Previous center-based interventions promoting physical activities and teacher-led activities have reported similar favorable impacts on gross motor development in young children in a childcare setting [52,55,56,57].

Additionally, the relationship between motor-skill competence and physical activity is reciprocal such that gross motor competence can inform ability to engage in physical activity and greater opportunities to be physically active can influence motor-skill competence [2,58]. ¡Míranos! offered support and encouragement at both the center and the child’s home to increase physical activity and reduce sedentary behaviors. These included strategies to break up prolonged sitting, reduce screen time, and increase PA through indoor-physical-activity breaks using small-space activities (i.e., GoNoodle), activities designed for home spaces, and family play time, contributing to gross motor development in both CBI and CBI + HBI children. ¡Míranos! enhanced children’s opportunities to be physically active and, thus, contributed to improvement in gross motor development.

Epidemiological and intervention studies have consistently demonstrated that a higher level of gross motor skills is predictive of a higher level of PA and improved health-related outcomes in childhood and later years [58,59,60]. Although participants in the intervention groups demonstrated significant improvement in gross motor skills, their gross motor development remained below the national norms for age and gender in both locomotive and ball skills at baseline and post test [21,61]. Studies showed that Latino children from low-income families are at higher risk of poor motor skill development [23,28] due to structural barriers such as limited access and opportunities to participate in physical activity which require an investment of time and financial resources from childcare providers and families [62,63]. Previous research also indicates that Latino parents who consider their neighborhood and physical environments to have low traffic risks, plenty of places for children to engage in PA, and adequate neighborhood informal social control may feel more comfortable engaging in practices that encourage their children to be physically active [64]. However, Latino families from low-SES backgrounds, such as the families in this study, are less likely to view their neighborhoods as safe and live near places where children can play [65,66]. Although our study was culturally tailored for Latinos from low socioeconomic backgrounds, little is known about cultural factors that influence gross motor development, let alone what strategies may be most culturally appropriate for teaching gross motors skills to Latino children [67]. Given that motor development is a prelude to cognitive and social–emotional development in young children [68,69], there is an urgent need for equity-oriented and culturally congruent physical-activity programs to reduce the health disparity in gross motor development in Latino children [70].

There are several limitations in this study that need to be considered when interpreting the findings. First, each component of ¡Míranos!, i.e., CBI and HBI, had multiple intervention strategies for multiple children behaviors, and therefore, it was not possible to ascertain which component or strategy had more influence on gross motor development. Furthermore, the study was not designed to compare the effect of the treatment conditions on gross motor development. Second, children’s gross motor skills were only assessed twice, i.e., once at baseline and once at post test, and only one evaluator was responsible for rating gross motor skills. Although this was carried out to reduce the burden of data collection on Head Start centers, it did not allow for the assessment of either inter- or intra-rater reliability. Third, although the TGMD-2 has consistently been found to be a valid and reliable measure of gross motor skills in young children across various races/ethnicities, previous research has noted that some children exhibit confusion when performing the skills in the sequence suggested by the TGMD-2 [71]. This may be because the order in which a child moves from one skill to the next during the TGMD-2 is not a sequence that would be reflected in natural play, which may impact children’s ability to perform the motor skill and adversely affect their scores.

Fourth, it is unclear why children in the CBI + HBI did not experience the same robust effect on BS skill scores as those in the CBI group since they used identical center-based strategies. Evaluation of intervention implementation fidelity showed that 100% of both CBI and CBI + HBI centers implemented ¡Míranos! PA and nutrition policies. Moreover, weekly reports from Head Start directors and staff indicated that 84% and 92% of intervention activities were implemented following the 26-week intervention schedule in CBI and CBI + HBI centers, respectively. Given the fidelity-related results, one would expect children in CBI centers to have BS skill scores worse than or equal to those in CBI + HBI centers, but this was not the case. However, quality of implementation was not assessed and may have impacted findings related to BS skill scores. For instance, one possibility is that the CBI centers had an instructional climate that was more conducive to the development of object-control skills. Previous research has indicated low-autonomy (i.e., teacher-centered) climates result in more proficient object-control skills among a disadvantaged preschool population compared to students in a comparison learning environment (i.e., unstructured recess) [72]. The ¡Míranos! physical activity/gross motor component included several teacher-led activities such as structured play and teacher-directed activity breaks during transition periods. Centers in the CBI group may have been more effective at implementing these teacher-led activities and fostering a low-autonomy climate, thereby enhancing children’s ball skills above and beyond children in the control condition. Finally, although the parent received education and training on modifications of the home environments to support physical activity and reduce sedentary behavior, ¡Míranos! intervention did not directly address the neighborhood environment or offer community resources for safe physical-activity opportunities that might have reduced the effectiveness of the HBI [73,74].

## 5. Conclusions

A comprehensive and culturally competent intervention targeting childcare centers and children’s homes was effective at improving children’s gross motor development and reducing disparities in child development. Given that gross motor development is a prelude to physical, cognitive, and social–emotional development in young children, the findings from this study provide clear evidence that child caregiver organizations should consider adopting policies and best practices to enhance GMS in young children.

## Figures and Tables

**Figure 1 ijerph-20-06974-f001:**
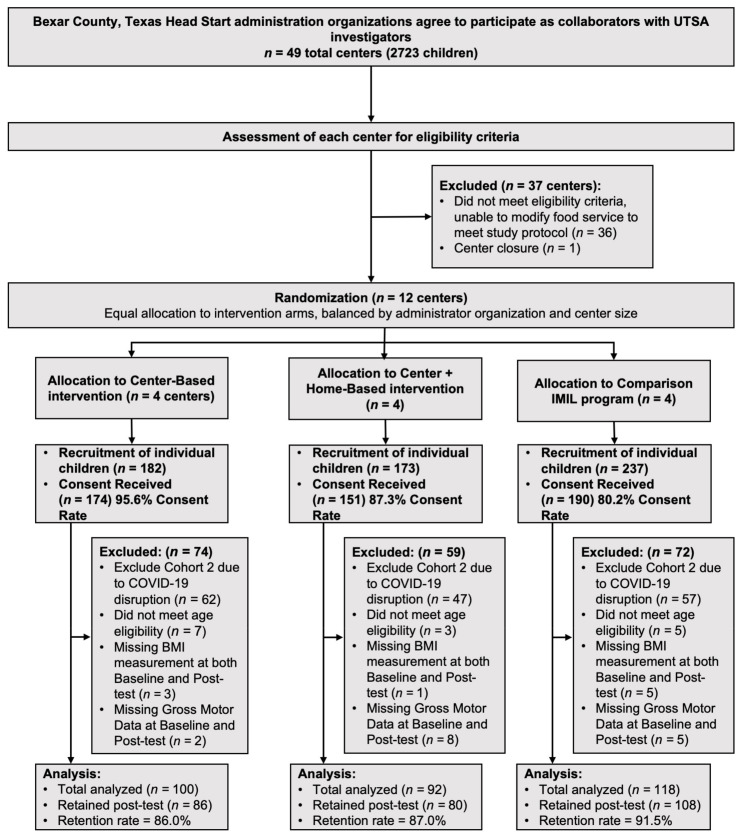
Data flow diagram.

**Figure 2 ijerph-20-06974-f002:**
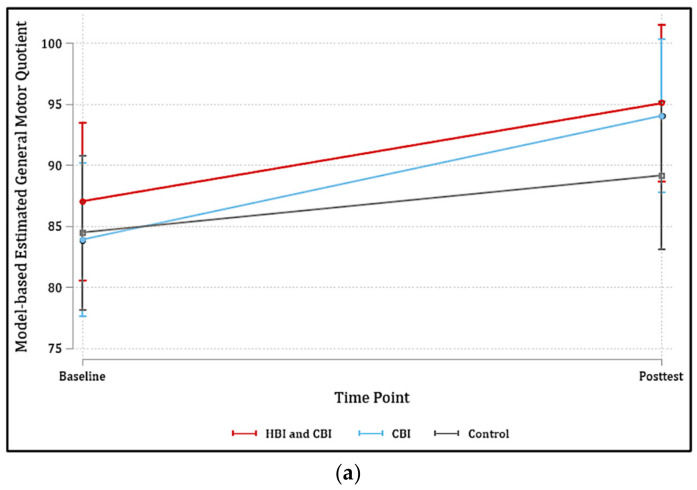
Scores of general motor quotient (**a**), locomotive skills percentiles (**b**), and ball skills percentiles (**c**) at baseline and post test by condition (i.e., HBI and CBI, CBI only, and Control).

**Table 1 ijerph-20-06974-t001:** Treatment conditions.

Treatment Condition	Treatment Component	Activities	Implementers
Center-based Intervention	Physical activity, nutrition policy and environment	Offer 90 min free and structured physical activity to children every day. Offer balanced healthy meals and snacks utilizing the USDA Child and Adult Care Food Program best practice recommendations.	Classroom teachers and paraprofessionals
	¡Míranos! physical activity/gross motor program	Children provided with 15 min of teacher-led activities using ¡Míranos! Activity Cards during outdoor play and then given 15 min of free play. Children participate in active-learning classroom activities during center time, transition, and breaks (30 min) each day. Children provided with outdoor play (structured activity 15 min and free play 15 min) each day. Screen time for entertainment at the center limited to 30 min a week. Children’s sitting time <15 min in any setting except nap and mealtime.	Classroom teachers and paraprofessionals
	Supplemental health-education activities	Use health-education activities from Sesame Street’s Healthy Habits for Life daily lessons. Provide weekly health challenges. Offer food tastings.	Classroom teachers and paraprofessionals
	¡Míranos! Staff-wellness program	Staff wellness challenges to promote healthy lifestyle changes.	All center staff
	Meal-pattern modification	Modify lunch and snacks offered to children to increase fresh fruit, vegetable, and whole-grain intake, reduce sugar intake, and increase exposure to healthy food.	Central kitchen staff
	Staff development and training	Review ¡Míranos! intervention plan and policies.	Center staff, curriculum development staff
Home-based Intervention	Peer-led obesity education	Monthly wall-poster parent-education sessions on eight health topics (15 min), to change parental obesity-related beliefs and practices, and teach parents recommendations on preschool children’s body weight, PA, nutrition.	Trained parent peer educators
	Home visits	Head Start staff conduct three home visits (30–45 min) with parents over the school year and help parents develop skills and strategies to promote PA, healthy eating, and sleep for their children at home.	Family-service workers
Comparison	“I Am Moving, I Am Learning” obesity-prevention curriculum	Promote physical activity and healthy eating. Keep active control centers engaged in study.	Control centers
	Literacy education	Teach skills for parents to introduce basic literacy and nutrition concepts at home. Keep parents engaged in study.	Control centers

**Table 2 ijerph-20-06974-t002:** Demographics and characteristics of Head Start centers and study participants.

Variables	CBI + HBI (*n* = 92)	CBI (*n* = 100)	Control (*n* = 118)	Total (*n* = 310)	*p* Value ^1^
Center size, *n* (%)					0.01
Small	57 (61.96)	71 (71)	61 (51.69)	189 (60.97)	
Large	35 (38.04)	29 (29)	57 (48.31)	121 (39.03)	
Child age at baseline, yr					0.49
Median (Q1, Q3)	3.64 (3.39, 3.84)	3.55 (3.36, 3.76)	3.61 (3.31, 3.92)	3.59 (3.33, 3.84)	
Mean ± SD	3.6 ± 0.29	3.56 ± 0.26	3.6 ± 0.32	3.59 ± 0.29	
Child sex, *n* (%)					0.18
Male	39 (42.39)	35 (35)	56 (47.46)	130 (41.94)	
Female	53 (57.61)	65 (65)	62 (52.54)	180 (58.06)	
Child race/ethnicity, *n* (%)					0.15
Non-H AA	5 (5.43)	10 (10)	4 (3.39)	19 (6.13)	
Hispanic	78 (84.78)	84 (84)	109 (92.37)	271 (87.42)	
Other	9 (9.78)	6 (6)	5 (4.24)	20 (6.45)	
Child with asthma, *n* (%)	8 (8.7)	15 (15)	17 (14.41)	40 (12.9)	0.35
Mother education, *n* (%)					0.7
Less than a high-school diploma	11 (11.96)	12 (12)	13 (11.02)	36 (11.61)	
High-school diploma/GED	42 (45.65)	38 (38)	54 (45.76)	134 (43.23)	
College or technical-school degree	27 (29.35)	41 (41)	40 (33.9)	108 (34.84)	
N/A or missing	12 (13.04)	9 (9)	11 (9.32)	32 (10.32)	
Language spoken most often at home, *n* (%)					0.63
English	52 (56.52)	62 (62)	58 (49.15)	172 (55.48)	
Spanish or other	22 (23.91)	22 (22)	30 (25.42)	74 (23.87)	
English and Spanish equally	12 (13.04)	11 (11)	21 (17.8)	44 (14.19)	
Not reported	6 (6.52)	5 (5)	9 (7.63)	20 (6.45)	
Parents not married, *n* (%)	34 (36.96)	35 (35)	44 (37.29)	113 (36.45)	0.93
Family members with history of diabetes, *n* (%)	41 (44.57)	43 (43)	43 (36.44)	127 (40.97)	0.44

CBI + HBI, Center- and home-based intervention; CBI, Center-based intervention. ^1^ *p* values are comparisons of the differences among the three groups.

**Table 3 ijerph-20-06974-t003:** Descriptive statistics of outcomes of interest by study groups.

Outcomes	CBI + HBI (*n* = 92)	CBI (*n* = 100)	Control (*n* = 118)	Total (*n* = 310)	*p* Value ^1^
GMQ					
Baseline ^2^	87.2 ± 9.22	83.74 ± 12.17	84.31 ± 12.35	85.01 ± 11.47	0.41
Posttest ^3^	95.09 ± 9.21	94.17 ± 9.46	89.25 ± 10.7	92.5 ± 10.21	<0.001 *
Change ^4^	8.92 ± 10.68	9.36 ± 13.04	4.33 ±1 4.06	7.3 ± 12.96	0.02 *
Cohen’s d	0.84 ^9^	0.72 ^9^	0.31 ^9^		
Cohen’s d	0.37 ^10^	0.37 ^11^			
Cohen’s d	−0.04 ^12^				
LS pctl					
Baseline ^5^	18.98 ± 15.34	23.78 ± 19.4	19.73 ± 17.1	20.73 ± 17.36	0.35
Posttest ^3^	42.59 ± 22.39	42.37 ± 22.35	30.25 ± 19.89	37.65 ± 22.17	<0.001 *
Change ^6^	25.13 ± 20.72	17.19 ± 23.92	10.2 ± 20.81	17.21 ± 22.54	<0.001 *
Cohen’s d	1.21 ^9^	0.72 ^9^	0.49 ^9^		
Cohen’s d	0.72 ^10^	0.31 ^11^			
Cohen’s d	0.35 ^12^				
BS pctl					
Baseline ^7^	35.95 ± 19.76	30.64 ± 19.16	34.67 ± 19.84	33.76 ± 19.65	0.13
Posttest ^3^	38.92 ± 19.84	37.36 ± 17.81	33.63 ± 19.73	36.35 ± 19.25	0.15
Change ^8^	4.14 ± 23.48	5.36 ± 20.93	−2.53 ± 24.7	1.97 ± 23.38	0.08
Cohen’s d	0.18 ^9^	0.26 ^9^	−0.1 ^9^		
Cohen’s d	0.28 ^10^	0.34 ^11^			
Cohen’s d	−0.05 ^12^				

CBI + HBI, Center- and home-based intervention; CBI, Center-based intervention; GMQ, General motor quotient; LS pctl, Locomotive skill percentile; BS pctl, Ball skill percentile; Change = post intervention− baseline; Entries are mean ± SD. * *p* < 0.05. ^1^ *p* values are comparisons of the differences among the three groups based on Kruskal–Wallis H test. ^2^ Sample sizes are 83, 88, and 100 in the HBI + CBI, CBI, and control groups, respectively. ^3^ Sample sizes are 80, 86, and 108 in the HBI + CBI, CBI, and control groups, respectively. ^4^ Sample sizes are 71, 74, and 90 in the HBI + CBI, CBI, and control groups, respectively. ^5^ Sample sizes are 83, 77, and 89 in the HBI + CBI, CBI, and control groups, respectively. ^6^ Sample sizes are 71, 65, and 80 in the HBI + CBI, CBI, and control groups, respectively. ^7^ Sample sizes are 84, 89, and 101 in the HBI + CBI, CBI, and control groups, respectively. ^8^ Sample sizes are 72, 75, and 91 in the HBI + CBI, CBI, and control groups, respectively. ^9^ Effect size for the pre–post-intervention difference (post test–baseline). ^10^ Effect size for the pre–post-intervention difference between CBI + HBI and control (CBI + HBI−control). ^11^ Effect size for the pre–post-intervention difference between CBI and control (CBI−control). ^12^ Effect size for the pre–post-intervention difference between CBI + HBI and CBI (CBI + HBI−CBI).

**Table 4 ijerph-20-06974-t004:** Adjusted change in outcomes of interest ^1^.

Outcomes	CBI + HBI (*n* = 92)	CBI (*n* = 100)	Control (*n* = 118)	Difference (CBI + HBI−Control)		Difference (CBI−Control)	
	Mean Change (SE)	Mean Change (SE)	Mean Change (SE)	Difference (95% CI)	*p* Value	Difference (95% CI)	*p* Value
GMQ ^2^	8.08 (1.48) *	10.15 (1.43) *	4.69 (1.32) *	3.39 (−0.50, 7.28)	0.09 ^	5.46 (1.64, 9.29)	0.01 *
LS pctl ^3^	23.99 (2.48) *	17.91 (2.53) *	10.31 (2.31) *	13.68 (7.04, 20.33)	<0.001 *	7.60 (0.89, 14.31)	0.03 *
BS pctl ^4^	3.32 (2.59)	6.11 (2.52) *	−1.46 (2.32)	4.78 (−2.04, 11.60)	0.17	7.57 (0.86, 14.28)	0.03 *

CBI + HBI, Center- and home-based intervention; CBI, Center-based intervention; GMQ, General motor quotient; LS pctl, Locomotive skill percentile; BS pctl, Ball skill percentile. ^ *p* < 0.1 * *p* ≤ 0.05. ^1^ All models take into account the correlations between multiple measures from the same child and multiple children from the same center and adjust for treatment, time, treatment×time, center size, and outcome-specific significant confounding variables as noted below. ^2^ Based on a linear mixed-effects model of 545 observations (average observations per child = 1.8, average children per center = 45.4) adjusting for mother’s education and language; ICC = 0.26 for measures nested within children; ICC = 0.06 for children nested within centers. ^3^ Based on a linear mixed-effects model of 523 observations (average observations per child = 1.7, average children per center = 43.6) adjusting for race/ethnicity, mother’s education and language; ICC = 0.36 for measures nested within children; ICC = 0.04 for children nested within centers. ^4^ Based on a linear mixed-effects model of 548 observations (average observations per child = 1.8, average children per center = 45.7) adjusting for baseline age and language; ICC = 0.28 for measures nested within children; ICC = 0.03 for children nested within centers.

## Data Availability

The datasets used and/or analyzed during the current study are available from the corresponding author on reasonable request.

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
