# Peer review of "¡Miranos! An 8-Month Comprehensive Preschool Obesity Prevention Program in Low-Income Latino Children: Effects on Children’s Gross Motor Development"

_ijerph, 2023, doi:10.3390/ijerph20216974_

Round 1
Reviewer 1 Report
Comments and Suggestions for Authors
General comments:
This is an interesting randomized controlled trial examining the effects of a comprehensive preschool obesity prevention program on motor development in low-income Latino children. The study is timely and addresses an important health disparity affecting Latino youth. The intervention strategy targeting preschool centers and homes is logical given the amount of time children spend in these settings. The findings contribute meaningful information to the literature on improving motor competence in early childhood as a way to impact long-term health. I have several suggestions to further strengthen the manuscript.
Introduction:
1. The introduction effectively establishes the importance of motor development in early childhood and the health disparities faced by Latino children. I would recommend adding 1-2 sentences on the link between obesity prevention and motor development to better justify how this intervention could influence motor skills.
2. Please provide a bit more detail on the Stodden model that is mentioned to explain the conceptual framework. For example, you could briefly describe the main components and assumptions of the model, and how they relate to your study.
Methods:
3. The description of randomized controlled trial methods and intervention arms is clear. Please provide more information on the sample - recruitment methods, eligibility criteria, enrollment rates, and retention rates. This will allow readers to better judge the generalizability of the findings.
4. Were there any measures taken to assess or control for intervention fidelity across sites? This could help explain the difference in ball skills results between CBI and CBI+HBI.
5. How do you handle missing data in your analysis? Did you use any imputation methods or conduct any sensitivity analysis?
Results:
6. The presentation of results is appropriate.
7. Did you conduct any power analysis or sample size calculations before conducting the study?
Discussion:
8. The authors did a good job interpreting the findings and comparing to past literature. The limitations are adequately acknowledged. Explore the limitations of the measures used - TGMD-2 provides standardized scores but has the children perform skills in a structured setting unlike natural play. How does this affect the validity and reliability of the measure?
9. More clearly address the unexpected ball skills findings and propose explanations. For example, you could discuss whether there were any differences in baseline characteristics, intervention exposure, or measurement error between CBI and CBI+HBI groups that could account for this result.
Conclusion:
10. The last sentence has a very strong expectation, I recommend revising the sentence as follows:
“Given that gross motor development is a prelude to physical, cognitive, and social-emotional development in young children, the findings from this study provide clear evidence that child caregiver organizations should consider adopting policies and best practices to enhance GMS in young children”
Reviewer 2 Report
Comments and Suggestions for Authors
First of all, I would like to congratulate you on your work. It is very thorough. I will therefore only mention small revisions which, in my opinion, will improve the understanding of the study.
- Line 45: Regarding references 6-8. I consider these to be too old. They should be replaced by more recent references. In this sense, I recommend the following manuscript: Schembri, R., Quinto, A., Aiello, F., Pignato, S., & Sgrò, F. (2019). The relationship between the practice of physical activity and sport and the level of motor competence in primary school children. Journal of Physical Education and Sport, 19, 1994-1998.
- Line 69: It might be useful to include some information about the reasons behind this phenomenon. For example, one reason is that it is not possible to access sports courses for a fee.
- Lines 70-73: For the revision of this section of the Introduction we recommend the following Manuscript: Hulteen, R. M., Morgan, P. J., Barnett, L. M., Stodden, D. F., & Lubans, D. R. (2018). Development of fundamental movement skills: A conceptual model for lifelong physical activity. Sports medicine, 48, 1533-1540.
- line 143: add the reason that led you not to choose the TGMD-3
- line 168: add the reason that led you to insert the age squared.
- lines 261-265: To affirm this, it would be good to explain the significance in the tables of descriptive statistics and, to understand the size of these differences. Therefore, it would be advisable to calculate ES by explaining it in the same table or by creating two distinct ones: one for the pre-post intervention differences and another for the differences between groups.
- lines 317-328: It should be made clear that another limitation is that of having had only one evaluator who, moreover, has made his assessments only once per phase (only one pre-intervention evaluation and only one post-intervention evaluation). This did not allow either inter-rater or intra-rater reliability to be assessed.
- It is necessary to rewrite the bibliography in accordance with the indications. The names of the journals should be abbreviated according to the ISO model.
Reviewer 3 Report
Comments and Suggestions for Authors
1. We haven't identified the purpose of the study, objectives, research question/questions.
2. After the eight months of the intervention, when the effectiveness in improving the motor development of the children included in the research is found, what happens to these children?
3. From our point of view, the conclusions should also highlight the importance of the fact that the families who signed the protocol also respected it.
We believe that the topic is relevant in the field, considering the constantly increasing number of obese people, especially children.I found the same type of information in an article by the same authors, published in 2022, in the present article considering the effects of the program on gross motor development.
¡Míranos! a comprehensive preschool obesity prevention programme in low-income Latino children: 1-year results of a clustered randomised controlled trial (cambridge.org)
Children's progress should be monitored after the program ends. It was, moreover, the second question formulated by us and addressed to the authors. Children's progress should be monitored after the program ends. It was, moreover, the second question formulated by us and addressed to the authors. References are appropriate?Author Response
Please see the attachment.

Reviewer 4 Report
Comments and Suggestions for Authors
This study examines the effects of the 8-month ¡Míranos! Intervention on children’s gross motor development. The study hypothesis was that children in the combined CBI and HBI or CBI-only conditions would have a significantly larger improvement in gross motor skills compared to control children at the posttest immediately after the completion of the intervention.
Introduction: No issues
Materials and Methods:
Include the ethics approval statement and number in the text.
Improve the appearance of Table 1 in the text. Make sure the text in the columns is aligned.
Improve the quality of Figure 1 and Figure 2.
In Table 3, ± missing spaces between them.
Overall, a good written study.
Comments on the Quality of English Language
Minor editing might be needed which could be picked up during proofreading of the manuscript.
